# Impact of phylogeny on the inference of functional sectors from protein sequence data

Nicola Dietler[1,2], Alia Abbara[1,2], Subham Choudhury[1,2], Anne-Florence Bitbol[1,2]*

**1** Institute of Bioengineering, School of Life Sciences, École Polytechnique Fédérale de Lausanne (EPFL), Lausanne, Switzerland, **2** SIB Swiss Institute of Bioinformatics, Lausanne, Switzerland

* anne-florence.bitbol@epfl.ch

**Data Availability Statement:** Our code is freely available at https://github.com/Bitbol-Lab/Phylogeny-sector-inference.

**Funding:** This project has received funding from the European Research Council (ERC) under the

## Abstract

Statistical analysis of multiple sequence alignments of homologous proteins has revealed groups of coevolving amino acids called sectors. These groups of amino-acid sites feature collective correlations in their amino-acid usage, and they are associated to functional properties. Modeling showed that nonlinear selection on an additive functional trait of a protein is generically expected to give rise to a functional sector. These modeling results motivated a principled method, called ICOD, which is designed to identify functional sectors, as well as mutational effects, from sequence data. However, a challenge for all methods aiming to identify sectors from multiple sequence alignments is that correlations in amino-acid usage can also arise from the mere fact that homologous sequences share common ancestry, i.e. from phylogeny. Here, we generate controlled synthetic data from a minimal model comprising both phylogeny and functional sectors. We use this data to dissect the impact of phylogeny on sector identification and on mutational effect inference by different methods. We find that ICOD is most robust to phylogeny, but that conservation is also quite robust. Next, we consider natural multiple sequence alignments of protein families for which deep mutational scan experimental data is available. We show that in this natural data, conservation and ICOD best identify sites with strong functional roles, in agreement with our results on synthetic data. Importantly, these two methods have different premises, since they respectively focus on conservation and on correlations. Thus, their joint use can reveal complementary information.

## Author summary

Proteins perform crucial functions in the cell. The biological function of a protein is encoded in its amino-acid sequence. Natural selection acts at the level of function, while mutations arise randomly on sequences. In alignments of sequences of homologous proteins, which share common ancestry and common function, the amino acid usages at different sites can be correlated due to functional constraints. In particular, groups of collectively correlated amino acids, termed sectors, tend to emerge due to selection on functional traits. However, correlations can also arise from the shared evolutionary history

European Union's Horizon 2020 research and innovation programme (grant agreement No. 851173, to A.-F. B.). The funders had no role in study design, data collection and analysis, decision to publish, or preparation of the manuscript.

**Competing interests:** We have no competing interests.

of homologous proteins, even without functional constraints. This may obscure the inference of functional sectors. By analyzing controlled synthetic data as well as natural protein sequence data, we show that two very different methods allow to identify sectors and mutational effects in a way that is most robust to phylogeny. We suggest that considering both of these methods allows a better identification of functionally important sites from protein sequences. These results have potential impact on the design of new functional sequences.

## Introduction

Statistical inference from genome and protein sequence data is currently making great progress. This is due both to the important growth of available genome sequences, and to the development of inference methods ranging from interpretable models inspired by statistical physics and information theory to large deep learning models. One important type of data sets consists in multiple sequence alignments (MSAs) of homologous proteins. These proteins, which are said to be the members of a protein family, share a common ancestry and common functional properties. Because of this, MSA columns (i.e. residue sites) feature correlations in amino-acid usage. Pairwise correlations due to contacts between amino acids in the three-dimensional structure of proteins have been thoroughly studied [1–23]. Statistical analyses of MSAs have also revealed the existence of groups of collectively correlated amino acids, termed sectors, which are associated to a functional role and are often spatially close in the protein structure [24–29]. Preserving the pairwise correlations of natural sequences in a sector-based analysis enabled the successful design of new functional sequences in a pioneering work [26]. Sectors can be identified using Statistical Coupling Analysis (SCA), which focuses on the large-eigenvalue modes of a covariance matrix weighted by conservation [27, 30]. It was recently shown in a minimal model that sectors of collectively correlated amino acids emerge from nonlinear selection on an additive trait (i.e. functional property) of the protein [31]. This case should be quite generic, since empirical data can often be modeled as nonlinear functions of a linear trait [32], and since many protein traits involve additive contributions from amino acids [32–34]. These modeling results motivated a principled method, called ICOD (for Inverse Covariance Off-Diagonal), which is designed to identify functional sectors, and more generally mutational effects, from sequence data [31]. Here, the focus is on the large-eigenvalue modes of the ICOD matrix, which essentially correspond to the small-eigenvalue modes of the covariance matrix, making ICOD *a priori* quite distinct from SCA. Another important difference is that ICOD was designed to focus on correlations and eliminate conservation signal as well as possible. Accordingly, a strength of ICOD is its robustness to conservation [31].

Because homologous sequences share a common ancestry, MSAs also comprise correlations coming from phylogeny [27, 35, 36]. These correlations can arise even in the absence of any selection, i.e. in the case where all mutations are neutral. Phylogenetic correlations impair the inference of structural contacts from sequences [4, 36–40], which has motivated empirical corrections aiming at reducing their impact [4, 7–9, 41–43, 43–45]. Disentangling them from collectively correlated groups of amino acids such as functional sectors is bound to be a significant challenge too, perhaps even more. Indeed, phylogeny strongly impacts the covariance matrix of sites of an MSA, and its particular its large-eigenvalue modes [35, 36], while sector identification by SCA focuses on the large-eigenvalue modes of a modified covariance matrix [27, 30]. Accordingly, the top mode identified in [27] was discarded for this reason,

and one of the sectors allowed to classify sequences by organism type, which suggests a phylogenetic contribution.

Here, we investigate how phylogeny impacts the inference of functional sectors, and more generally, of mutational effects, from protein sequence data. Fully disentangling correlations from phylogeny and from functional constraints in natural data is challenging. Thus, we propose a minimal model comprising both phylogeny and functional sectors, allowing us to generate synthetic data with controlled amounts of phylogeny and of functional constraints. We use this data to quantify the impact of phylogeny on sector identification and on mutational effect inference by SCA and ICOD, but also by conservation and correlation. We find that ICOD is most robust to phylogeny, but that conservation is also quite robust. Next, we consider MSAs of 30 natural protein families for which deep mutational scan experiments have been performed, yielding measurements of mutational effects on a specific aspect of function [46]. We show that conservation and ICOD best identify sites with strong functional roles, consistently with our results on synthetic data. Importantly, these two methods have different premises, since they respectively focus on conservation and on correlations.

## Results

### Minimal model of sequences with a functional sector and phylogeny

**Functional sector.** We consider sequences where each site can take two states, $-1$ and $1$. While our model can be generalized to include 20 states, reflecting the number of different amino acids, restricting to two states allows to retain key features within a minimal model. We focus on a scalar trait, which corresponds to a physical property associated to a given protein function. Examples of traits include the binding activity to a ligand and the catalytic activity of an enzyme. The trait is assumed to be additive: for a given sequence $\vec{\sigma} = (\sigma_1, \ldots, \sigma_L)$ with length $L$, it reads $\tau(\vec{\sigma}) = \sum_i D_i \sigma_i$ where $\vec{D}$ is the vector of mutational effects. Thus, each site is assumed to contribute independently to the trait, with its state impacting the trait value.

Nonlinear selection on such an additive scalar trait was shown to lead to a sector [31], i.e. a collectively correlated group of amino acids [27]. The sector corresponds to a specific group of functional amino acids in the protein that are particularly important for the trait of interest. In our model, sector sites have a large mutational effect on the trait. Following [31], we assume that a specific value $\tau^*$ of the trait is favored by natural selection. Selection is thus performed using the quadratic Hamiltonian

$$H(\vec{\sigma}) = \frac{\kappa}{2} \left( \sum_{i=1}^{L} D_i \sigma_i - \tau^* \right)^2, \tag{1}$$

where $\kappa$ denotes selection strength. Note that the fitness of a sequence is minus the value of the Hamiltonian. Equilibrium sequences under the Hamiltonian in Eq 1 can be sampled using the Metropolis-Hastings algorithm, see Fig 1A. The resulting data sets contain only correlations coming from selection on the sector.

**Phylogeny.** To incorporate phylogeny, we take an equilibrium sequence and we evolve this ancestral sequence along a perfect binary tree, see Fig 1B. Along the tree, random mutations changing the state of an amino acids are proposed, and they are accepted or rejected according to the same Metropolis criterion as the one used to generate equilibrium sequences. Thus, selection on the trait is maintained through the phylogeny. A fixed number $\mu$ of accepted mutations are performed on each branch. As a result, two closest (resp. farthest) sequences on the leaves of a tree differ at most by $2\mu$ mutations (resp. $2\mu n$ mutations, where $n$ is the number of generations, i.e. of branching events, in the tree). Note that these numbers are reduced if

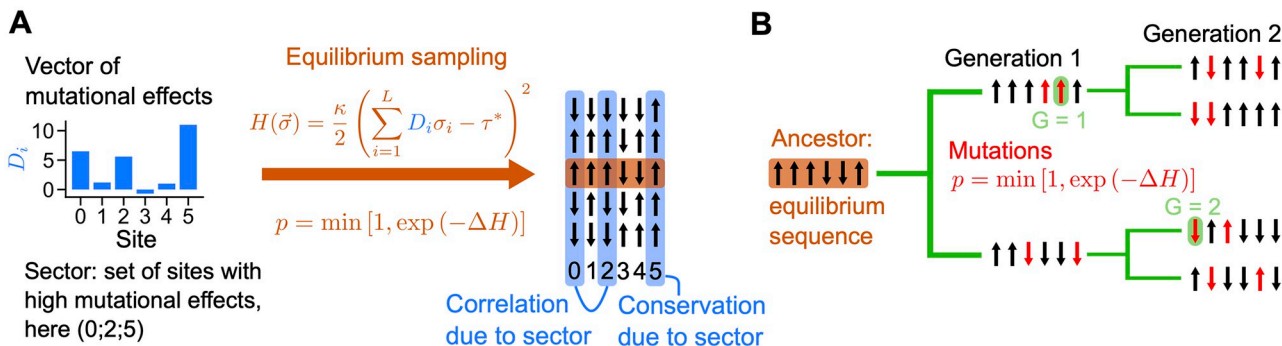

**Fig 1. Data generation process. A**: Generation of sequences with selection only. Given a vector of mutational effects $\vec{D}$ on the trait of interest, we sample independent equilibrium sequences under the Hamiltonian $H(\vec{\sigma})$ in Eq 1. For this, we start from random sequences and we use a Metropolis Monte Carlo algorithm where proposed mutations (changes of state at a randomly chosen site) are accepted with probability $p$, according to the Metropolis criterion associated to $H(\vec{\sigma})$. The obtained sequences feature pairwise correlations and conservation arising from selection on the trait (via the Hamiltonian $H(\vec{\sigma})$ in Eq 1). **B:** To incorporate phylogenetic correlations, we start from one equilibrium sequence, which becomes the ancestor. We evolve it on a perfect binary tree over a fixed number $n$ of "generations" (i.e., tree levels, corresponding to branching events), here $n = 2$ generations. A fixed number of mutations $\mu$ (red, here $\mu = 2$) are accepted with probability $p$ on each branch of the tree. The earliest generation at which a site mutates with respect to its ancestral state is denoted by $G$, see examples highlighted in green.

multiple substitutions occur at the same site. Generated sequences at the leaves of the tree therefore contain correlations from phylogeny and from selection. Their importance are controlled respectively by the parameters $\mu$ and $\kappa$: a smaller $\mu$ means that sequences are more closely related, yielding more phylogenetic correlations, while a larger $\kappa$ means stronger selection, yielding more correlations arising from the sector. Note that this data generation process with phylogeny is close to the one we used previously [40, 47], but that the Hamiltonian we use here (Eq 1) is specific to the sector model.

## Some previous results on sector identification

**SCA.** Sectors were first defined in natural data by using Statistical Coupling Analysis (SCA), a method which detects groups of collectively correlated and conserved sites in sequences [27, 30]. In SCA, sectors correspond to the sites that have large components in the eigenvectors associated to the largest eigenvalues of the covariance matrix of sites in the sequences weighted by conservation.

**ICOD.** Another method to detect sectors was introduced in Ref. [31], and focuses on the eigenvectors with large eigenvalues of the inverse covariance matrix of sites, with diagonal terms set to zero. This method is called ICOD, for Inverse Covariance Off-Diagonal. The eigenvalues of the ICOD matrix are thus close to the inverse of those of the covariance matrix of sites, and large ICOD eigenvalues map to small eigenvalues of the covariance matrix. Hence, ICOD and SCA focus on a different part of the spectrum of the covariance matrix. Besides, ICOD does not use conservation weighting, and further removes conservation signal by setting diagonal terms to zero. It thus focuses on correlations, while SCA combines conservation and correlations. Conceptually, the Hamiltonian in Eq 1 entails that selected sequences satisfy $\vec{D} \cdot \vec{\sigma} \approx \tau^*$. All selected sequences have a similar projection on the vector $\vec{D}$ of mutational effects and lie close to a plane orthogonal to $\vec{D}$. Therefore, $\vec{D}$ is a direction of particularly small variance, which is why the eigenvector with smallest eigenvalue of the covariance matrix, and the one with largest eigenvalue of the ICOD matrix, contains information on $\vec{D}$ and on the sector [31].

## Impact of selection and phylogeny on ICOD, covariance and SCA spectra

**Signatures of sectors in the spectrum of the ICOD matrix.**   Before studying the effect of phylogeny, let us analyze the ICOD matrix and its spectrum without phylogeny, which will serve as baseline. To first order in $\kappa$ (small selection regime), the off-diagonal elements of the ICOD matrix can be approximated as $\tilde{C}_{ij}^{-1} \approx \kappa D_i D_j$ for all $i, j$ between 1 and $L$ with $i \neq j$ [31]. Here, we show using two different methods that this formula still holds beyond first order in $\kappa$: one method extents this formula to second order, and the second one to fourth order (see S1 Appendix section 1). Within this robust approximation, if diagonal elements were equal to $\kappa D_i^2$ instead of 0, the spectrum of the ICOD matrix would comprise a single nonzero eigenvalue $\kappa \sum_{i=1}^{L} D_i^2$ associated to the mutational effect vector $\vec{D}$. Hence, the eigenvector associated to the largest eigenvalue of the ICOD matrix allows to recover $\vec{D}$, as observed in [31].

Consider a mutational effect vector $\vec{D}$ comprising $L_S$ sites with large effects, corresponding to the functional sector, and $L - L_S$ sites with substantially smaller effects. Reordering sites to group together first the sector sites and then the non-sector sites yields an ICOD matrix containing one block of size $L_S \times L_S$ corresponding to the sector, and another of size $(L - L_S) \times (L - L_S)$ which encompasses other sites with low mutational effects. To gain insight on the ICOD spectrum of this matrix, let us ignore the elements that do not belong to either of these blocks by setting them to 0 (see S1 Fig). The modified matrix is a block diagonal matrix, and its eigenvalues are the union of the eigenvalues of each block. Apart from its zero diagonal elements, each block is well-approximated by an outer product matrix with elements $\kappa D_i D_j$ whose absolute value is large for the sector block, and small for the non-sector block. If diagonal elements were equal to $\kappa D_i^2$ instead of 0, the spectrum of the sector (resp. non-sector) block matrix would comprise a single nonzero eigenvalue $\kappa \sum_{i=1}^{L_S} D_i^2$ (resp. $\kappa \sum_{i=L_S+1}^{L} D_i^2$). Setting the diagonal elements to zero entails that eigenvalues in each block need to sum to 0 (as the trace is 0), which leads to some perturbation of the nonzero eigenvalue, but moreover to the appearance of negative eigenvalues that compensate this large positive one. These negative eigenvalues are expected to be larger in absolute value in the sector block than in the non-sector block.

Fig 2 illustrates that the spectrum of the ICOD matrix is reasonably well approximated by that of its block diagonal approximation, if the ICOD matrix is obtained over many sequences. Furthermore, in the block diagonal approximation, we observe that eigenvalues from the sector block yield the largest eigenvalue and the smallest (most negative) eigenvalues, while the non-sector block provides the intermediate eigenvalues. This is in agreement with our expectations coming from the outer product approximation of these blocks. Thus, in addition to possessing a largest eigenvalue coming from the sector and associated to an eigenvector close to the mutational effect vector $\vec{D}$, the ICOD matrix also comprises smallest (most negative) eigenvalues that contain information on the sector.

Note that in the example shown in Fig 2, important mutational effects corresponding to the sector are all positive. We checked that our results are robust to the case where the sector comprises both sites with large positive and and large negative mutational effects: as shown in S2 Fig, the spectrum is then very similar to that in Fig 2. Furthermore, the top eigenvector of the ICOD matrix successfully recovers $\vec{D}$.

**Impact of phylogeny on ICOD, covariance and SCA spectra.**   How do phylogenetic correlations affect the signature of sectors in the spectra of the ICOD, covariance and SCA matrices? To answer this question, we generate data with only selection, or only phylogeny, or both, within our minimal model.

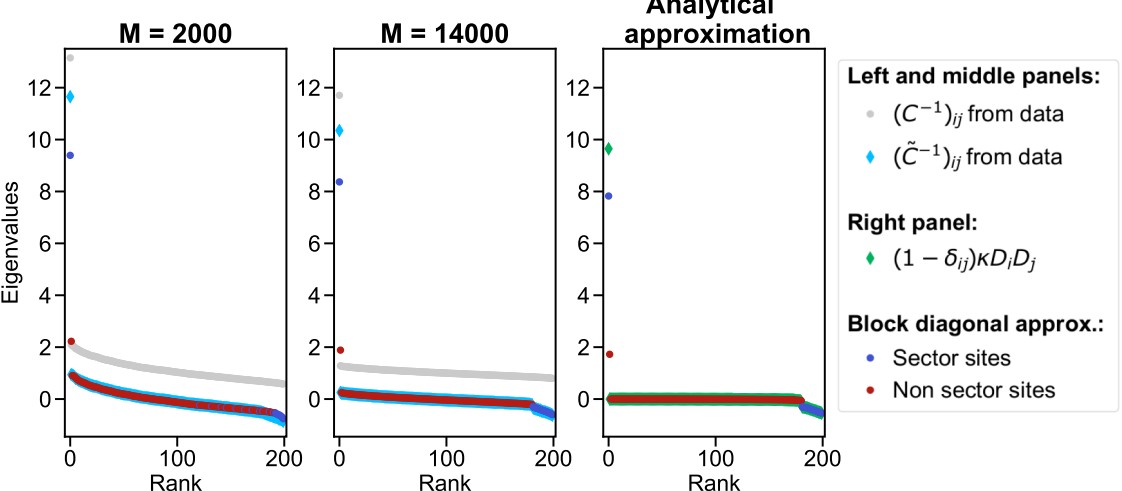

**Fig 2. Spectrum of the ICOD matrix and of its block diagonal approximation.** Left (resp. middle) panel: spectrum of the ICOD matrix $\tilde{C}^{-1}$ computed on 2000 (resp. 14,000) sequences generated independently at equilibrium, and of its block diagonal approximation. The spectrum of the inverse covariance matrix $C^{-1}$ is also shown as a reference. Right panel: spectrum of the analytical approximation of the ICOD matrix, and of its block diagonal approximation. Sequences of length $L = 200$ were sampled independently at equilibrium using the Hamiltonian in Eq 1 with $\kappa = 10/(\sum_i D_i^2)$ and $\tau^* = 90$. The vector of mutational effect $\vec{D}$ comprises sector sites (the 20 first sites) with components sampled from a Gaussian distribution with mean 5 and variance 0.25, and non-sector sites (the remaining 180 sites) with components sampled from a Gaussian distribution with mean 0.5 and variance 0.25. The analytical approximation $\tilde{C}_{ij}^{-1} \approx (1 - \delta_{ij})\kappa D_i D_j$ (see S1 Appendix section 1) was computed from the values of $\kappa$ and $\vec{D}$ used for data generation.

Fig 3 shows the spectra of the ICOD, covariance and SCA matrices for these data sets. Without phylogeny, a sector gives rise to one large eigenvalue (rank 0) that is an outlier in the spectrum of the ICOD matrix [31]. We observe that this outlier is preserved even with strong phylogeny (small $\mu$, here $\mu = 5$), although its contrast with the rest of the spectrum decreases when $\mu$ decreases. Comparing to baseline data without selection confirms that this outlier is due to selection, and also shows that some signal associated to selection is present in the small eigenvalues of the ICOD matrix (rank close to 199), in agreement with our analysis above. This effect of selection also remains visible with phylogeny.

Recall that because of the matrix inversion in ICOD, large ICOD eigenvalues essentially map to small covariance eigenvalues. Accordingly, comparing data with and without selection shows that the main signature of selection is observed for the smallest eigenvalue of the covariance matrix (rank 199). While this outlier is strong without phylogeny, its contrast with the rest of the spectrum decreases as phylogeny is increased, and it is no longer a clear outlier for $\mu = 5$. Meanwhile, the large eigenvalues of the covariance matrix are strongly impacted by phylogeny, in agreement with previous findings [36], but little impacted by selection.

Finally, the large eigenvalues (rank close to 0) of the SCA matrix contain selection signal, as they are outliers with selection, and not without. However, these outliers disappear when phylogeny is strong ($\mu = 5$), and the eigenvalues with and without selection are then very similar. Interestingly, except in the high-phylogeny case, the number of large-eigenvalue outliers in SCA matches the number of sector sites in $\vec{D}$.

Thus, overall, the signatures of selection in the spectrum of ICOD appear to be more robust to phylogeny than those observed in the spectra of the covariance matrix and of the SCA matrix.

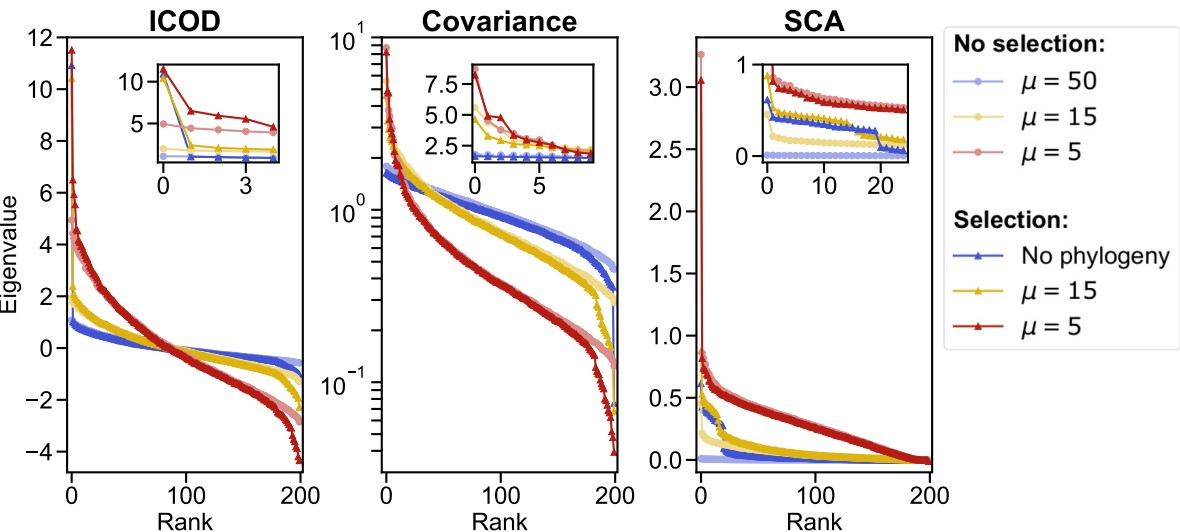

**Fig 3. Impact of phylogeny and selection on ICOD, covariance and SCA spectra.** Eigenvalues of the ICOD, covariance and SCA matrices, sorted from largest to smallest, are shown for sequences generated with only phylogeny (light shades) and both phylogeny and selection (dark shades). We consider different levels of phylogeny by considering different values of $\mu$ (shown as different colors). 'No phylogeny' corresponds to sequences generated independently at equilibrium, and thus containing only correlations due to selection. This data set comprises $M = 2048$ sequences of length $L = 200$ generated exactly as in Fig 2, i.e. using the Hamiltonian in Eq 1 with $\kappa = 10/(\sum_i D_i^2)$, $\tau^* = 90$, and the same vector of mutational effect $\vec{D}$ as in Fig 2. Data sets without selection are generated by evolving random sequences of length $L = 200$ on a perfect binary branching with 11 generations and $\mu$ random mutations on each branch, providing $M = 2^{11} = 2048$ sequences. Finally, data sets with phylogeny and selection are generated along a perfect binary tree with $\mu$ accepted mutations per branch (with acceptance criterion in Eq 2 using the same $\kappa$ and $\tau^*$ as in the no-phylogeny case and as in Fig 2) and 11 generations again. The three values of $\mu$ shown here were chosen to illustrate different levels of phylogenetic impact. Insets show a zoom over large eigenvalues. A logarithmic y-scale is used in the center panel for readability.

## Impact of phylogeny on mutational effect recovery

How much signal from the sector and the mutational effect vector $\vec{D}$ is contained in the eigenvectors of the ICOD, covariance and SCA matrices? How is this impacted by phylogeny? To quantitatively address this question, we employ the recovery score (see Methods), which is the normalized scalar product of the eigenvector of interest and of the vector $\vec{D}$, both of them with their components replaced by their absolute values. The absolute value is used because our main goal is to identify sites with large mutational effect, whatever its sign, and because eigenvectors are defined up to a global sign.

We consider the eigenvectors associated to the main outliers identified in Fig 3, i.e. the largest eigenvalue for ICOD and SCA, and the smallest one for covariance. We also consider conservation as a baseline, since it is known to be efficient at identifying sites under selection in natural protein sequence data (see e.g. [48]). How well do these different eigenvectors, or conservation, recover $\vec{D}$ when varying the amount of phylogeny? Fig 4 shows the recovery versus the number of mutations per branch $\mu$ for ICOD, SCA, covariance, and conservation. We observe that ICOD is more robust than covariance and SCA to phylogeny, and performs much better than covariance in strong phylogenetic regimes (small $\mu$). ICOD and covariance have recoveries close to the maximum possible value of 1 for intermediate to weak phylogeny (larger $\mu$), meaning that they recover almost perfectly $\vec{D}$. Still for large $\mu$, SCA and conservation reach similar performance, which is less good than that of ICOD and covariance. Furthermore, ICOD outperforms conservation, except for very small $\mu$ (very strong phylogeny). Conservation performs better than covariance and SCA with strong phylogeny, while the opposite holds

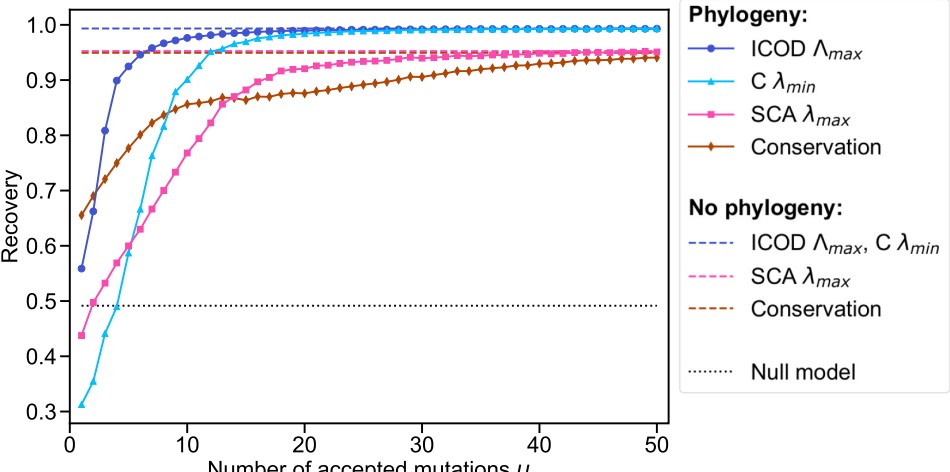

**Fig 4. Impact of phylogeny on mutational effect recovery.** The recovery of the mutational effect vector $\vec{D}$ (see Methods) for specific eigenvectors is shown as a function of the number $\mu$ of mutations per branch, using ICOD, covariance, SCA and conservation. For ICOD (resp. SCA), eigenvectors associated to the largest eigenvalue $\Lambda_{max}$ (resp. $\lambda_{max}$) are considered. For covariance $C$, eigenvectors associated to the smallest eigenvalue $\lambda_{min}$ are considered. Datasets of $M = 2048$ sequences of length $L = 200$ were generated along a perfect binary tree with 11 generations, using various numbers $\mu$ of accepted mutations per branch. As in Fig 3, we employed the mutation acceptance criterion in Eq 2 with $\kappa = 10/(\sum_i D_i^2)$ and $\tau^* = 90$. We used the same vector of mutational effect $\vec{D}$ as in Fig 2 and Fig 3. All results are averaged over 100 realisations of data generation. The null model corresponds to recovery from a random vector (see Methods, Eq 13).

with weaker phylogeny. In Fig 4 and throughout, we employ a small pseudocount for ICOD but not for other approaches (see Methods). This is motivated by the need to invert the covariance matrix in ICOD [31]. For covariance, the difference in the recovery between results with and without pseudocount (taking the same pseudocount value as for ICOD) is lower than 1.2% for all values of $\mu$. For SCA, it is lower than 0.3%, and for conservation it is lower than $3 \times 10^{-4}$%. Thus, the pseudocount does not affect the conclusions.

As we showed above that signal about selection also exists at the other end of the spectrum, we investigate it too in S3 Fig. The resulting performance is substantially worse than in Fig 4. For ICOD and covariance, this is expected from our theoretical analysis, since it is the eigenvector with smallest variance that should be close to $\vec{D}$ (see above and [31]). Nevertheless, we note that the eigenvector associated to the smallest ICOD eigenvalue $\Lambda_{min}$ also contains information about $\vec{D}$. Indeed, S3 Fig shows that it outperforms the null model corresponding to recovery by a random vector (see Methods and [31]).

While we focused on the largest and smallest eigenvalues because they are expected to contain most of the signal about the sector, it is interesting to look at the recovery of all eigenvectors. S4 Fig confirms that the best recoveries (closest to 1) are found in the eigenvector associated to the largest (rank 0) eigenvalue for ICOD and SCA, and to the smallest one (rank 199) for covariance. However, these recoveries decrease with increased phylogeny (i.e. for smaller $\mu$), but ICOD is most robust, consistent with our previous results. For ICOD, we further observe relatively large recovery scores for $L_S - 1$ small eigenvalues (here $L_S = 20$, so the associated ranks are 180–199), in agreement with our formal analysis of the spectrum and our expectation that smallest eigenvalues should regard the sector. S4 Fig shows that this holds both without and with phylogeny.

So far, we discussed the impact of phylogeny on mutational effect recovery for fixed values of the favored trait value $\tau^*$ and of the selection strength $\kappa$. These two parameters, which

characterize selection, impact recovery too [31]. We study their interplay with phylogeny in section 2 of the S1 Appendix.

**Generalization to other phylogenetic trees.** How does the type of phylogenetic tree affect the performance of sector inference? So far, and in particular in Fig 4, synthetic data was generated using a perfect binary tree for simplicity (see Methods). However, natural phylogenies are more complex. To assess the impact of more general phylogenetic trees on our results, we generate synthetic data using Beta-coalescent trees, following Ref. [49], instead of perfect binary trees. These trees range from a Bolthausen-Snitzman coalescent (with selection) for a Beta-coalescent tree with $\alpha = 1$ to a Kingman coalescent (neutral case) for a Beta-coalescent tree with $\alpha = 2$, see examples in S5 Fig. In S6 Fig, we show the impact of branch length on mutational effect recovery, as in Fig 4, but for the trees shown in S5 Fig. Throughout S6 Fig, we find similar results regarding the relative performance of ICOD, conservation and covariance as in Fig 4. Furthermore, our results regarding the performance of SCA compared to other methods are also similar to those in Fig 4 for $\alpha = 1$ and $\alpha = 1.2$. For $\alpha = 1.4$ and $\alpha = 1.6$, we observe that, in one tree realization out of two, SCA has lower performance with strong phylogeny than for other trees, and features a slower convergence to its no-phylogeny asymptote. Conversely, for $\alpha = 1.8$ and $\alpha = 2$, we find that SCA performs better than in other cases, and reaches performances similar or slightly better than those of other methods. These results suggest that SCA is more sensitive than other methods to the details of a phylogenetic tree, and may be particularly useful for neutral or quasi-neutral cases. Apart from these points, our main results are robust to considering diverse phylogenetic trees.

## Impact of phylogenetic correlations

How does phylogeny impact the components of the key eigenvectors of the ICOD, covariance and SCA matrices? To investigate this question, we generate data using our minimal model and keeping track of all intermediate ancestral sequences in the phylogenetic tree. Next, we assign to each site of the sequence a score $G$, which corresponds to the earliest generation, i.e., tree level, at which the site mutates, and thus becomes different from the initial state in the ancestral sequence, see Fig 1B. In Fig 5, we examine how the eigenvector components relate to $G$, for two data sets, generated with two different values of the number $\mu$ of mutations per branch of the tree. These values yield respectively small and large effects of phylogeny on the mutational effect recovery in Fig 4, and match two of the three values shown in Fig 3. The mean pairwise Hamming distances in these synthetic datasets are 0.47 for $\mu = 50$ and 0.30 for $\mu = 5$. They fall in the range observed in natural data, see S1 and S2 Tables. Fig 5 shows that in the weak phylogeny regime, ICOD, covariance and SCA all yield much larger eigenvector components for sector sites than for other sites. Conversely, with strong phylogeny, the eigenvector components associated with sector and non-sector sites strongly overlap when using covariance and SCA. Remarkably, ICOD is still able to disentangle them. We also note that covariance scores are strongly affected by $G$: sites that mutate late have stronger scores, presumably because they are quite conserved and thus feature little variance, resulting in a high contribution to the eigenvector associated with the smallest eigenvalue of the covariance matrix. Recall that while the eigenvector associated to the largest ICOD eigenvalue also focuses on small variances, the diagonal of the ICOD matrix is set to 0 in order to remove conservation effects. Consistently, the contributions of sites to the ICOD eigenvector only increase moderately with $G$. Furthermore, S7 Fig considers the eigenvector associated with the largest eigenvalue of the covariance matrix (other end of the spectrum) under strong phylogeny. There, sites that mutate early in the phylogeny obtain high scores. Thus, the spectrum of the covariance matrix is strongly impacted by phylogeny. Similarly, SCA attributes higher scores to sites

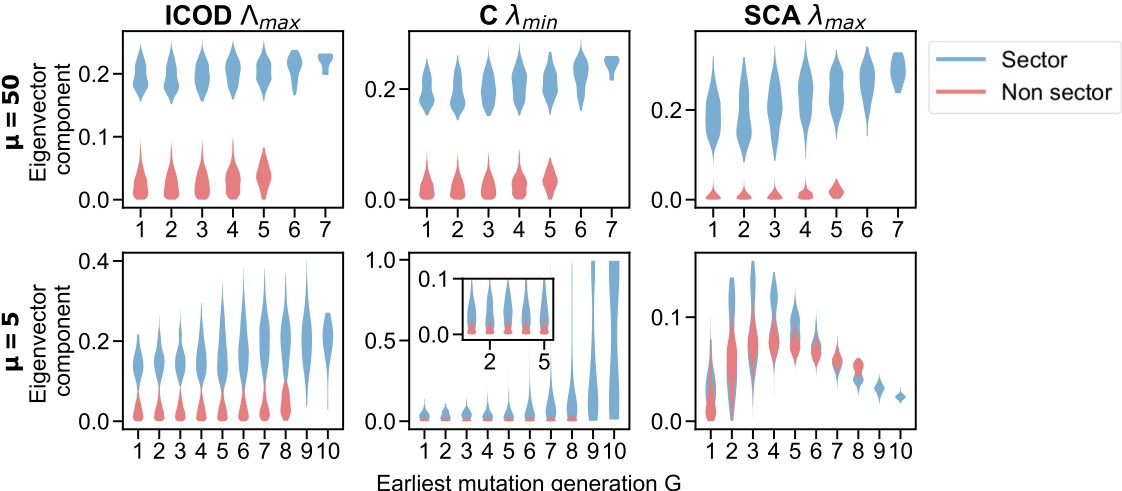

**Fig 5. Impact of earliest mutation generation *G* on eigenvector components.** Violin plots of the absolute value of components of the key eigenvectors of the ICOD, covariance *C* and SCA matrices are represented versus the earliest mutation generation *G* at which the associated site first mutates in the phylogeny. Results are shown for data sets generated with $\mu = 50$ (top panels) and $\mu = 5$ (bottom panels). Datasets of $M = 2048$ sequences of length $L = 200$ were generated along a perfect binary tree with 11 generations, using two different numbers $\mu$ of accepted mutations per branch. As in Fig 3, we employed the mutation acceptance criterion in Eq 2 with $\kappa = 10/(\sum_i D_i^2)$ and $\tau^* = 90$. We used the same vector of mutational effect $\vec{D}$ as in Figs 2 and 3. Violin plots are obtained over 100 realisations of data generation.

that mutate relatively early in the phylogeny and might be less conserved, leading to a higher variance and high contributions to the eigenvector associated with the largest eigenvalue of the SCA matrix.

Fig 5 further shows that sector sites tend to mutate later (higher *G* score) than other sites. Indeed, selection impedes mutation of sites with high mutational effects, as these mutations tend to substantially deteriorate fitness.

## Identifying functionally important sites in natural protein families

How does ICOD perform at predicting sites with large mutational effects on natural data? Does its robustness to phylogeny give it an advantage? To address this, we consider natural protein families with published Deep Mutational Scan (DMS) experimental data, which we consider as ground truth. We investigate 30 protein families [46] listed in S1 Table. For each of them, we rank sites by the magnitude of predicted mutational effects using ICOD, SCA, Mutual Information (MI, see [45]), and conservation. We compare these predictions to DMS data. Specifically, for each protein family, we start from the reference sequence mutated in the DMS experiment, and construct a multiple sequence alignment of homologs of this protein whose depth can be varied by selecting the neighbors of the reference sequence in Jukes-Cantor distance up to a given phylogenetic cutoff [45]. Next, for each method, we compute the top eigenvector on MSAs constructed using different phylogenetic cutoffs, and then we summed component by component the eigenvectors corresponding to these different cutoffs. We use the components of the resulting vector as predictors of mutational effects, and the DMS data as ground truth. Note that there are other ways of aggregating scores across phylogenetic cutoffs. Averaging performance metrics across cutoffs gives results that are overall consistent with those obtained by summing eigenvector components, but slightly less good, see S8 Fig. Fig 6 shows that all methods are able to predict important sites, the most successful ones being

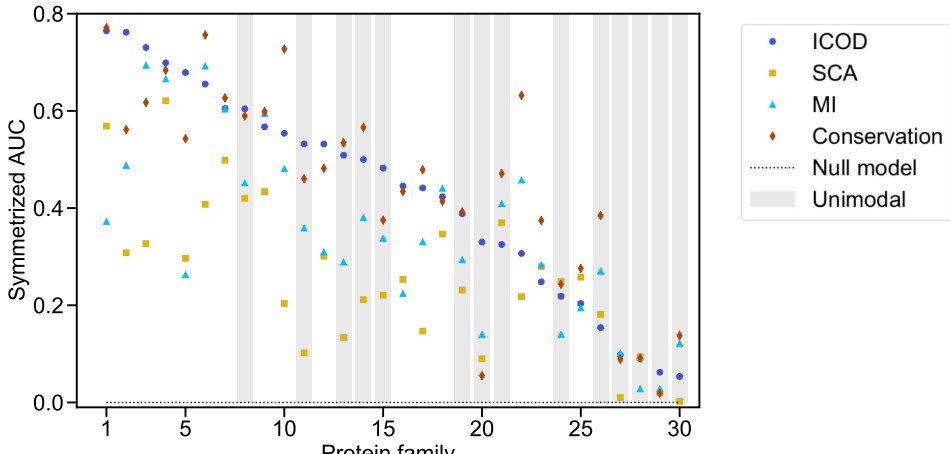

**Fig 6. Identifying functionally important sites in natural protein families.** The symmetrized AUC for the prediction of sites with large mutational effects is computed on 30 protein families, using four different methods: ICOD, SCA, MI and Conservation, using Deep Mutational Scan (DMS) data as ground truth. For ICOD and MI, the average product correction (APC) [4] is applied to the matrix of interest (it was found to improve the average performance for most families for these methods, but not for SCA). For ICOD, MI and SCA, the components of the eigenvector associated to the largest eigenvalue are employed to make predictions of mutational effects. Protein families are ordered by decreasing symmetrized AUC for ICOD. The mapping between protein family number and name is given in S1 Table. The protein families shaded in grey have DMS data featuring a unimodal shape, the other ones have a bimodal shape.

conservation and ICOD. Indeed, the average of the symmetrized AUC over all families is 0.45 for conservation, 0.43 for ICOD, 0.35 for MI and 0.26 for SCA. Furthermore, we observe that better results tend to be obtained for DMS with bimodal distributions of fitness effects (see S9 Fig for examples). This suggests that when the data comprises sites with substantially stronger mutational effects than others for the function probed by the DMS (i.e. a functional sector), then this sector can be well identified by the methods considered here, in particular conservation and ICOD.

The observation that ICOD and conservation tend to outperform SCA is consistent with our results obtained in synthetic data. To assess more directly the link between these results and phylogeny in natural data, S10 Fig shows the relative difference of ICOD and SCA performances versus MSA diversity, characterized by the mean pairwise Hamming distance between sequences in the MSA (see also S2 Table). We find that for all 7 families with mean pairwise Hamming distance smaller than 0.55, ICOD significantly outperforms SCA, while results are more mixed for more diverse families. These 7 least diverse families are precisely those for which we expect the effect of phylogeny to be strongest. This result suggests that phylogeny does explain part of the success of ICOD on these natural datasets.

We note that in many cases, focusing on specific cutoffs yielded better performance than the sum of eigenvectors over cutoffs considered here. However, optimal cutoffs differed between families and methods, making it difficult to exploit this for prediction. For completeness, S11 Fig shows the best performance obtained across cutoffs. The average of the symmetrized AUC over all families is 0.47 for conservation and for ICOD, 0.42 for MI and 0.37 for SCA. Thus, our conclusion that ICOD and conservation perform best is robust to considering the best cutoff instead of summing over cutoffs.

While conservation and ICOD yield the best identification of functionally important sites, they rely on entirely different signals. Indeed, ICOD was designed to focus primarily on correlations and not on conservation—recall that the diagonal of the inverse covariance matrix is set to zero in order to eliminate conservation. Accordingly, with synthetic data, we

observe in Fig II in section 2 of the S1 Appendix that ICOD reaches very good performance for a favored trait value $\tau^* = 0$, such that conservation is uninformative. Thus, we expect these two methods to find different important sites. S1 Table shows the number of sites identified as functionally important by both ICOD and conservation ("Both") or only by either method ("ICOD" or "Cons."), out of those that are deemed important by the DMS, i.e. the $L_S$ sector sites found by binarising the DMS data and setting a cutoff (see Methods). There is often substantial overlap between sites found by ICOD and by conservation, which suggests that in practice many important sites are both somewhat conserved and somewhat correlated with others. Nevertheless, we find that the predictions of ICOD and of conservation can substantially differ, in line with our expectations. For instance, $\beta$-lactamase has 12 sites that are predicted by ICOD only and 8 sites by conservation only, although the two methods yield a similar symmetrized AUC. We examined the 3D structures of HRas and PDZ and the location of the sites identified by both methods or by only one method in S12 and S13 Figs. In these two cases, DMS data is obtained by focusing on ability to bind other molecules. Accordingly, functionally important sites identified in these DMS are located around interaction regions. We observe that the sites detected by ICOD only tend to be closer to these interfaces than those detected by conservation only. These sites have large DMS scores and are localized close to binding interfaces, strongly hinting at their functional importance, while they are not among the most conserved ones. Interestingly, ICOD allows to reveal such sites in these examples.

## Discussion

It was shown in [31] that nonlinear selection acting on any additive functional trait of a protein gives rise to a functional sector in the sequence data associated to the protein family of interest. While the main signature of the selection process lies in the small-eigenvalue modes of the covariance matrix of sites, motivating ICOD, the large-eigenvalue modes were nevertheless affected [31]. Here, we showed that this arises from the mathematical properties of the covariance matrix provided that some sites have much larger mutational effects than others. We further generalized the analytical approximation of the elements of the ICOD matrix, thereby reinforcing the theoretical bases of ICOD.

Next, using synthetic data from a minimal model where we can tune the amounts of phylogeny and of nonlinear selection on an additive trait, we showed that phylogenetic correlations can substantially impair the inference of functional sectors from sequence data. However, ICOD and conservation are generally more robust to phylogenetic noise than SCA and covariance. The robustness of ICOD suggests that focusing on the small-eigenvalue modes of the covariance matrix is a successful strategy, as the large-eigenvalue ones tend to be most affected by phylogeny [35, 36]. Conservation has been used a lot to identify functionally important sites (see e.g. [48]), and thus, its good performance is not surprising. In addition, it was argued in [50] that at least in some cases, the success of SCA can be attributed to its use of conservation. Here, we find that SCA's performance is indeed often tied to that of conservation. Importantly, in ICOD, the diagonal terms of the inverse covariance matrix are set to zero in order to eliminate the impact of conservation as much as possible, and focus on correlations. Thus, the robustness of ICOD cannot be explained by that of conservation, and the two methods should provide different and complementary information. While SCA aimed to combine the two important and complementary ingredients of conservation and covariance, our analysis of controlled synthetic data suggests that considering the two separate ingredients and using ICOD instead of covariance to further separate them may yield even more information.

Our analysis of several natural protein families for which DMS data is available showed that conservation and ICOD were the best predictors of mutational effects among the methods we considered. This could be due to the robustness of these methods to phylogenetic correlations, especially given that ICOD particularly outperforms SCA for natural protein families with small diversity, where phylogenetic correlations should be most important. Here, we showed that phylogenetic correlations make the inference of functional sectors challenging, very much like they obscure the inference of structural contacts [4, 36–40]. It is important to note that phylogenetic correlations are nevertheless interesting and provide useful signal e.g. for the inference of protein partners among paralogs [47, 51, 52].

Our model of selection on an additive trait with a quadratic Hamiltonian is formally close to Potts models [7–23] and to generalized Hopfield models [53, 54]. Potts models have allowed mutational effect analysis [46, 55–57]. However, our selection model induces a specific dependence of the fields and of the couplings in the mutational effect vector which does not generically exist in these models. This is what makes it possible to recover mutational effects directly via an eigenvector of the ICOD matrix, instead of needing to compare inferred energies [56]. Methods able to predict mutational effects can be used to determine sites with top mutational effects. We compared using the ICOD eigenvector and using the Potts model energy for this as in [46], and found that the latter had slightly better average symmetrized AUC. Note however that the results of [46] were obtained using different MSAs, which makes the comparison imperfect. We stress that eigenvector-based methods, such as SCA and ICOD, have the potential to disentangle different aspects of function via different eigenvectors [27, 30, 31], which is not the case of Potts energy-based methods. Studying the impact of phylogeny on cases with simultaneous selection on multiple traits, as in Ref. [31], would be interesting. Furthermore, generalizing beyond quadratic models would be very interesting [58, 59]. Besides, while we started from a generic model for sectors which involves an additive trait [32–34] under nonlinear natural selection, it would be interesting to analyze specific traits in more detail. One of them [31] is the energetic cost of protein deformations within an elastic-network model [60]. The low-energy deformation modes of protein structures are important in many functionally important protein deformations [61–65], and are robust to sequence variation within protein families [66–68]. It would be interesting to reexamine these cases with a more in-depth analysis of phylogenetic effects.

Our work provides a step towards understanding the impact of the rich structure of biological data on the performance of inference methods [69]. We focused on traditional and principled inference methods, because these interpretable methods allow us to get direct insight into the origin of method performance. However, we believe that the insight gained can also be useful for deep learning approaches. In particular, the question of disentangling purely phylogenetic signal from functional signal is important for all methods. Furthermore, many current deep learning models start from the same data structure as the methods considered here, namely MSAs. For instance, AlphaFold, which has brought major advances to the computational prediction of protein structures from sequences, starts by constructing an MSA of homologs when given a protein sequence as input, and its performance is strongly impacted by MSA properties [70]. Closer to our analysis, DeepSequence [46] employs a variational autoencoder trained on MSAs to predict mutational effects, and reaches strong performance. While protein language models relying on non-aligned protein sequences also allow successful mutational effect prediction [71], their performance is strongly impacted by the number of homologs that a sequence possesses, and increases with it [72], hinting that homologs remain important in these methods.

## Methods

### Generating synthetic sequences

To assess the impact of phylogeny on the prediction of mutational effects sectors, we generate sequences using our minimal model, either without phylogeny or with various levels of phylogeny.

**Generating independent equilibrium sequences.** A Metropolis Monte Carlo algorithm is used to sample equilibrium and independent sequences according to the Hamiltonian in Eq 1, see Fig 1A. To generate each sequence, we start from a random sequence and let it evolve by accepting a fixed number of mutations (spin flips), chosen large enough for sequences to equilibrate—in practice, 3000, which yields convergence of the energy of sequences. The probability $p$ that a proposed mutation is accepted is given by the Metropolis criterion

$$p = \min[1, \exp(-\Delta H)], \qquad (2)$$

where $\Delta H$ is the difference of the value of the Hamiltonian in Eq 1 with and without the proposed mutation.

In this way, we generate sequences sampled from the distribution $P(\vec{\sigma}) = \exp(-H(\vec{\sigma}))/\sum_{\vec{\sigma}} \exp(-H(\vec{\sigma}))$. The resulting sequences have trait values distributed around a favored value $\tau^*$, and the selection strength $\kappa$ regulates how much the trait values deviate from the favored value.

**Generating sequences with phylogeny.** In order to incorporate phylogeny, we take a functional sequence generated as above as the ancestor, and let it evolve on a perfect binary tree, see Fig 1B. A fixed number $\mu$ of accepted mutations are performed independently along each branch, and the sequences sequences at the tree leaves constitute our MSA. Mutations are accepted with probability $p$ from Eq 2, which maintains the same selection on the trait. The resulting sequences contain correlations from phylogeny (controlled by $\mu$) in addition to those coming from selection. If $\mu$ is large enough, even sister sequences become independent and equilibrium statistics are recovered. Conversely, if $\mu$ is small, sister sequences are very similar and phylogeny is strong. Note that we can also generate sequences with phylogeny and no selection using the same method, but accepting all proposed mutations.

### Natural data

**Comparing to experiments.** Deep Mutational Scan (DMS) experiments provide a direct measurement of the fitness effect associated to each possible point mutation at each site of a reference sequence. In DMS experiments, selection for a given function (e.g. ability to bind to another molecule) is applied on the mutated and the wild type sequences. The ratio of the number of sequences after and before selection is called $r_{WT}$ for the wild-type and $r_{mut}$ for the mutant. Most often, the fitness effect of the particular mutation is assessed via the logarithm of the enrichment ratio $\log(r_{mut}/r_{WT})$. Functionally important sites are those for which mutations are most costly, leading to a small $r_{mut}$ and a negative log enrichment score. Thus, we construct the analogue of our vector $\vec{D}$ of mutational effects by taking for each site the most negative (smallest) enrichment ratio, associated with the most deleterious mutation at this site. We start from the 30 DMS datasets collected in [46], listed in S1 Table, and with metrics describing their diversity and depth given in S2 Table. To define the set of sites with large mutational effects (which corresponds to the sector), we use a threshold on the DMS measurements to binarise them, leading sector and non-sector sites. The threshold is set as follows. We observe that most score distributions are in practice either bimodal or unimodal, see S9 Fig for some examples. Thus, we fit for each family either a linear combination of two Gaussian

distributions or a single Gaussian distribution to find a cutoff value for the binarisation. For the unimodal families, we use the mean of the fitted Gaussian distribution as the cutoff, while for the bimodal ones we use the position of the minimum between the two peaks. In some cases, there are more than two peaks in the distribution: then, we fit a bimodal distribution to the two most negative peaks, see upper middle panel in S9 Fig. In all cases, we define the sector as comprising the sites with values more negative than the cutoff value. The number of thus-defined sector sites is called $L_S$. Once the sector and non-sector sites are defined, the performance of various models at identifying sector sites is quantified by the symmetrized AUC. See the paragraph on "Performance evaluation" below for details.

**MSA construction.**   In order to infer sectors in natural data, we need to construct Multiple Sequence Alignments (MSAs) of homologs of the reference sequence of the DMS experiment. For this, we use the method from reference [46]. We search the reference sequence against the UniRef100 database using `jackhmmer` from the HMMER3 suite [73], specifically the command "`jackhmmer -incdomT 0.5*L -cpu 6 -N 5 -A pathsave pathrefseq pathuniref`", where $L$ is the length of the wild type sequence. We restrict to match states where the reference sequence does not have a gap, remove columns with more than 30% gaps, and remove sequences which have more than 20% gaps. Inspired by reference [45], we further subsample the MSA to generate a collection of MSAs comprising neighbors of the reference sequence up to different phylogenetic cutoffs [45]. Specifically, we only retain sequences up to a given maximum Jukes-Cantor distance [74], which we refer to as "phylogenetic cutoff", of the reference sequence. Each gap of the MSA is then replaced by the amino acid possessed at the same site by the nearest sequence in terms of Jukes-Cantor distance. We employ the following values of cutoff distance: 0.4, 0.6, 0.8, 0.9, 1.0, 1.1, 1.2, 1.3, 1.4, 1.5, 1.6, 1.7, 1.8, 1.9 and 2.0, yielding 15 MSAs for each protein family. Note that for protein families 1, 3, and 27, MSAs were generated only up to cutoffs 1.2, 1.4 and 1.4 respectively, due to computational constraints.

## Inferring mutational effects

In an MSA, each column is a protein site and each row is a sequence. All methods considered here rely on single-site and two-site frequencies of amino acids. Single-site frequencies are denoted by $f_i(\sigma_i)$ for a given state (or amino acid) $\sigma_i$ on the $i$th site of the protein. Similarly, two-site frequencies are denoted by $f_{ij}(\sigma_i, \sigma_j)$. The state $\sigma_i$ can take values (−1, 1) for synthetic data and (0, . . ., 19) for natural data, representing the 20 natural amino acids. To avoid divergences due to states that do not appear in the data for ICOD or Mutual information, a pseudo-count $a$ is introduced [7–9], leading to pseudocount-corrected frequencies

$$\tilde{f}_i(\sigma_i) = a/q + (1 - a)f_i(\sigma_i)\,, \tag{3}$$

$$\tilde{f}_{ij}(\sigma_i, \sigma_j) = a/q^2 + (1 - a)f_{ij}(\sigma_i, \sigma_j) \quad \text{for} \quad i \neq j\,, \tag{4}$$

$$\tilde{f}_{ii}(\sigma_i, \sigma_j) = \delta_{\sigma_i\sigma_j}\tilde{f}_i(\sigma_i)\,, \tag{5}$$

where $q$ is the number of states, i.e. $q = 2$ for synthetic data and $q = 20$ for natural data.

In the case of synthetic data, we apply each method (ICOD, covariance, SCA, conservation) to the generated MSA. In the case of natural data, we have 15 different MSAs for each protein family. We apply each method (ICOD, MI, SCA, conservation) on each MSA and extract the eigenvector associated to the eigenvalue of focus. This yields 15 different eigenvectors that are added together component by component. Because normalized eigenvectors are defined up to

an overall sign, we set the sign of one eigenvector arbitrarily and choose the one of the other eigenvectors such that they have a positive Pearson correlation with the first eigenvector. The overall sign of the sum remains arbitrary, but we employ scoring methods that are invariant to it. Note that for natural data, we do not aim to infer mutational effect values but only their rank, in line with the usual practice of SCA [27, 29]. While in principle we can infer mutational effect values from eigenvectors at least with ICOD [31], the measured fitness can involve additional nonlinearities not captured by our simple model.

**Conservation.**   We use an entropy-based definition of conservation [50]. For a given site $i$ ($i$th column in the MSA), it reads

$$\text{Conservation}_i = \log_q (q) - \sum_{\sigma_i} f_i(\sigma_i) \log_q (f_i(\sigma_i)) = 1 - \sum_{\sigma_i} f_i(\sigma_i) \log_q (f_i(\sigma_i)) \;, \qquad (6)$$

where $q$ is the number of states and $\log_q$ denotes the logarithm of base $q$. Specifically, this conservation measure is the Kullback-Leibler divergence of the amino acid frequencies with respect to the uniform distribution, which is taken as reference for simplicity (another possibility would be to use the background amino-acid frequencies [50]). The conservation score for each site $i$ can directly be compared to the mutational effect vector $\vec{D}$ for synthetic data, or to the DMS score for natural data.

**Covariance.**   The elements of the covariance matrix of sites can be calculated as

$$C_{ij}(\sigma_i, \sigma_j) = f_{ij}(\sigma_i, \sigma_j) - f_i(\sigma_i) f_j(\sigma_j) \;. \qquad (7)$$

Note that for ICOD, we use the pseudocount-corrected frequencies here (see above).

For synthetic data, in the two state case $(-1, 1)$ with $q = 2$, we use for covariance and ICOD the standard covariance definition

$$C_{ij} = \langle \sigma_i \sigma_j \rangle - \langle \sigma_i \rangle \langle \sigma_j \rangle \;, \qquad (8)$$

where $\langle \cdot \rangle$ denotes a mean across all sequences of the MSA [31]. Employing it is equivalent to using Eq 7 and restricting to one state, considering the other as reference, which can be done because normalization entails that the second state gives redundant information: $f_i(1) + f_i(-1) = 1$ [31]. For ICOD, the covariance matrix is computed with the pseudocount-corrected frequencies (see Eqs 3–5), and thus its elements read $C_{ij}^{(a)} = (1 - a)\langle \sigma_i \sigma_j \rangle - (1 - a)^2 \langle \sigma_i \rangle \langle \sigma_j \rangle$ for $i \neq j$ and $C_{ii}^{(a)} = (1 - a)^2 (1 - \langle \sigma_i \rangle^2) + a(2 - a)$. Meanwhile, for SCA, the full covariance matrix with all states (i.e. Eq 7) is used (see below).

For natural data, $C$ is a $20 \times L$ by $20 \times L$ matrix where for each pair of sites $i, j$ there is a sub-matrix of size $20 \times 20$ comprising each possible pair of amino acids. The Frobenius norm of each of these sub-matrices weighted by conservation is used in SCA (see [30] and below), while in ICOD we use the reference-sequence gauge and eliminate one redundant state (see below).

**ICOD.**   ICOD is a method introduced in [31], where the covariance matrix of sites is inverted (which requires using a pseudocount), and its diagonal terms are all set to zero. The latter allows to mitigate the impact of conservation [31]. Using the mean-field approximation of Potts model inference [7–9] it was shown in [31] that in the two-state case, starting from the covariance matrix in Eq 8, the ICOD matrix can be approximated as

$$(\tilde{C}^{-1})_{ij} \approx (1 - \delta_{ij}) \kappa D_i D_j \;. \qquad (9)$$

In practice, we use a pseudocount value $a = 1 \times 10^{-5}$ for synthetic data.

For natural data, a similar construction can be made by employing the covariance matrix in Eq 7 with pseudocount-corrected frequencies (Eqs 3 and 4), with a pseudocount value $a = 0.05$. The reference-sequence gauge [31] is used, meaning that the reference sequence of the dataset (i.e., the wild-type sequence for deep mutational scan data) is chosen as our baseline. Here, in practice, we do not compute the frequency of the amino acid in the reference sequence at a given position, yielding a covariance matrix of size $(19 \times L, 19 \times L)$. This matrix is then inverted, the Frobenius norm is taken on each submatrix of size $19 \times 19$ corresponding to each pair of sites $(i, j)$. Next, the diagonal is set to zero.

**SCA.** Statistical Coupling Analysis (SCA), introduced in [27, 30, 75], is a method that combines covariance and conservation. The elements of the SCA matrix are defined as

$$\tilde{C}_{ij}^{SCA}(\sigma_i, \sigma_j) = \phi_i(\sigma_i)\phi_j(\sigma_j)C_{ij}(\sigma_i, \sigma_j) , \tag{10}$$

where $C_{ij}(\sigma_i, \sigma_j)$ is an element of the full covariance matrix (Eq 7), while $\phi_i(\sigma_i)$ characterizes amino conservation on site $i$. The Frobenius norm of each $20 \times 20$ block of this matrix, which corresponds to a pair of sites $i, j$, is then computed, and it is on this matrix that we perform an eigendecomposition. More details on this method can be found in [30], and we use the corresponding implementation, namely the `pySCA` github package (https://github.com/reynoldsk/pySCA). For the natural data, we use default parameters [30]. For the synthetic case, we transformed our data from $(-1, 1)$ states to $(0, 1)$ states, and in `pySCA` we changed the number of states from 20 to 2 and the background frequency to 0.5 for each state.

**Mutual Information (MI).** The MI of each pair of sites $(i, j)$ is computed as

$$\text{MI}_{ij} = \sum_{\sigma_i, \sigma_j} \tilde{f}_{ij}(\sigma_i, \sigma_j) \log\left(\frac{\tilde{f}_{ij}(\sigma_i, \sigma_j)}{\tilde{f}_i(\sigma_i)\tilde{f}_j(\sigma_j)}\right), \tag{11}$$

where $\tilde{f}_i(\sigma_i)$ and $\tilde{f}_{ij}(\sigma_i, \sigma_j)$ are the pseudocount-corrected one- and two-body frequencies (see Eq 3–5). We set the pseudocount value to $a = 0.001$. Note that we use frequencies instead of probabilities to estimate mutual information, and do not correct for finite-size effects [76]. This is acceptable because we only compare scores computed on data sets with a given size, affected by the same finite size effects.

## Performance evaluation

In order to quantify how well methods perform, we use the recovery score introduced in [31]. The recovery of the vector $\vec{D}$ of mutational effects by any vector $\vec{v}$ is defined as

$$\text{Recovery} = \frac{\sum_i |v_i D_i|}{\sqrt{\sum_i v_i^2}\sqrt{\sum_i D_i^2}} , \tag{12}$$

The chance expectation of the recovery, corresponding to recovery by a random vector, is [31]

$$\langle\text{Recovery}\rangle \approx \sqrt{\frac{2}{\pi L}} \frac{\sum_i |D_i|}{\sqrt{\sum_i D_i^2}} . \tag{13}$$

We also evaluated performance using the AUC, i.e. area under the receiver operating characteristic. More precisely, we used the symmetrized AUC, defined as $2(|\text{AUC} - 0.5|)$. This allows us to account for the fact that eigenvectors are defined up to an overall sign.

For synthetic data, we mainly use the recovery score because the eigenvectors are likely to match the full vector $\vec{D}$. Indeed, there is only selection on one function with a quadratic nonlinearity and it is well captured by our methods. The symmetrized AUC is also studied in some

cases and there $\vec{D}$ is binarised accordingly to sites belonging to the sector or not. In general, the symmetrized AUC focuses on the ranking of important sites (sector sites), while recovery focuses on the similarity between the eigenvector considered and $\vec{D}$.

For natural data, we mainly use the symmetrized AUC to evaluate performance, because DMS only test one aspect of function and the measured fitness can involve additional nonlinearities not captured by our simple model. To calculate the AUC, we vary the threshold on the numbers of sites predicted (which are the top eigenvector components). Specifically, computing the true positive and false positive rates for each threshold value allows constructing the receiver operating characteristic curve and computing the AUC.

## Supporting information

**S1 Appendix. The Supplementary Appendix comprises calculations for the analytical approximation of the inverse covariance matrix at higher orders (section 1) and an analysis of the impact of selection parameters on mutational effect recovery (section 2).**
(PDF)

**S1 Fig. ICOD matrix and its block diagonal approximation.** The left panel shows the ICOD matrix computed on data generated independently at equilibrium (14000 sequences). Parameters are the same as for the equilibrium ('No phylogeny') data set in Fig 3. Recall that $L_S = 20$ sector sites out of $L = 200$ total sites. The first $20 \times 20$ diagonal block (mainly blue) is associated to the sector, while the second one, of size $180 \times 180$, is associated to non-sector sites (i.e. sites with small mutational effects). The right panel shows the block diagonal approximation of the ICOD matrix shown in the left panel. Here, the matrix elements that do not belong to either of the two diagonal blocks are set to 0. Meanwhile, elements of these diagonal blocks are the same as in the ICOD matrix.
(PDF)

**S2 Fig. Spectrum of the ICOD matrix and of its block diagonal approximation.** Same as in Fig 2, except that the vector $\vec{D}$ of mutational effects comprises both negative and positive components. Specifically, we took the same $\vec{D}$ as in Fig 2, but we multiplied the 10 first components (corresponding to half of the sector) by $-1$.
(PDF)

**S3 Fig. Impact of phylogeny on mutational effect recovery by the opposite end of the spectrum.** Same as in Fig 4, but using the eigenvectors associated to the eigenvalues at the opposite end of the spectrum. For ICOD (resp. SCA), eigenvectors associated to the smallest eigenvalue $\Lambda_{min}$ (resp. $\lambda_{min}$) are considered. For covariance $C$, the eigenvector associated to the largest eigenvalue $\lambda_{max}$ is considered.
(PDF)

**S4 Fig. Mutational effect recovery by each eigenvector.** The mutational effect recovery is shown for three methods: ICOD, Covariance and SCA, using each eigenvectors in the spectrum of the matrix involved in the method. The dashed line shows the chance expectation for the recovery (see Eq 13). Recovery by the eigenvectors associated to the largest and smallest eigenvalues are indicated for each method. Throughout, three data sets are considered, which differ by the amount of phylogeny (No phylogeny, $\mu = 15$, $\mu = 5$). The data is generated as in Fig 3.
(PDF)

**S5 Fig. Phylogenetic trees in the Beta-coalescent process.** Beta-coalescent trees are generated using the code associated to Ref. [49], which is available on GitHub (https://github.com/rneher/betatree). Each column corresponds to a specific value of the parameter $\alpha$ which characterizes the Beta-coalescent tree. For each value of $\alpha$, we sampled two trees, represented in the two rows of the figure. Note that $\alpha = 1$ corresponds to the Bolthausen-Sznitman coalescent, while $\alpha = 2$ corresponds to the Kingman coalescent.
(PDF)

**S6 Fig. Impact of phylogeny on mutational effect recovery for different phylogenetic trees.** Same analysis as in Fig 4, but using data generated along the Beta-coalescent trees shown in S5 Fig instead of a perfect binary tree. Each panel here is associated to the tree shown in the corresponding panel of S5 Fig. Mutational effect recovery is shown for various methods versus the average number $\langle m \rangle$ of accepted mutations between the ancestor and a leaf. For synthetic data generation, the number of accepted mutations along each branch is drawn from a Poisson law with mean equal to a multiplicative constant times the branch length shown in S5 Fig. The multiplicative constant (which is the same for all branches in a tree) is then varied to tune the amount of phylogeny in the data, yielding various values of $\langle m \rangle$. All curves are averaged over 100 realisations of data generation along the same tree. We checked that the asymptotic values obtained without phylogeny (dashed lines) are reached for large $\langle m \rangle$.
(PDF)

**S7 Fig. Impact of earliest mutation generation $G$ on covariance eigenvector components.** Same as in Fig 5, restricting to the data set with $\mu = 5$ and to the covariance method. Top panel: eigenvector associated to the largest eigenvalue $\lambda_{max}$ of the covariance matrix. Bottom panel: results for the eigenvector associated to the smallest eigenvalue $\lambda_{min}$ (see Fig 5) are reproduced here for comparison purposes.
(PDF)

**S8 Fig. Identifying functionally important sites in natural protein families: Mean performance over phylogenetic cutoffs.** Same as in Fig 6, but using the mean of the performance, i.e. mean of the symmetrized AUCs, over all phylogenetic cutoffs considered. The error bars represent the standard deviation over phylogenetic cutoffs. Here, the average of the symmetrized AUC over all families is 0.42 for conservation, 0.39 for ICOD, 0.25 for MI and 0.19 for SCA. The mapping between protein family number and name is given in S1 Table.
(PDF)

**S9 Fig. Distribution of DMS scores for 6 example protein families.** Histograms of the minimum DMS scores (see Methods) are shown in six example cases. Top panels: three examples of scores with bimodal shape (or more complex shape—middle panel, see Methods). Bottom panels: three examples of scores with unimodal shape. In each case, bimodal Gaussian (in top panels) or Gaussian fits (in bottom panels) are shown together with their respective cutoffs, which correspond either to the location of the minimum value between peaks for bimodal fits, or to the mean for Gaussian fits (see Methods).
(PDF)

**S10 Fig. Relative difference of ICOD and SCA performances versus MSA diversity.** The relative difference between the symmetrized AUC score $S$ for ICOD and SCA (i.e., ($S_{ICOD}$ − $S_{SCA}$)/$S_{ICOD}$) is plotted versus the mean pairwise Hamming distance in the MSA, for the prediction of sites with large mutational effects in 30 different protein families. The red dashed line separates the cases where ICOD is best (positive values) from those where SCA is best (negative values). Performances measured as symmetrized AUC values are the same as in Fig

6, and protein families are listed in S1 Table. For each family, the mean pairwise Hamming distance is computed for the MSA associated to the maximum phylogenetic cutoff $C_{max}$ considered, thus reflecting the largest diversity in the family, see S2 Table. Phylogenetic cutoffs are defined and their values are given in the paragraph "MSA construction" of the Methods section.
(PDF)

**S11 Fig. Identifying functionally important sites in natural protein families: Maximum performance over phylogenetic cutoffs.** Same as in Fig 6, but focusing on the MSA phylogenetic cutoffs that maximize the symmetrized AUC. The mapping between protein family number and name is given in S1 Table.
(PDF)

**S12 Fig. Sector of the PDZ domain inferred by ICOD and by conservation.** The structure of the PDZ domain is represented in cyan and the CRIPT molecule is shown in orange. True positive sites of the sector found both by ICOD and by conservation are colored in yellow, while TP sites found by ICOD only are shown in green and TP sites found by conservation only are shown in red. The PDB identifier of this structure is 1BE9.
(PDF)

**S13 Fig. Sector of the Ras protein inferred by ICOD and by conservation.** In both panels, the structure of Ras is shown in cyan. True positive sites of the sector found both by ICOD and by conservation are colored in yellow, while TP sites found by ICOD only are shown in green and TP sites found by conservation only are shown in red. (a) Structure of Ras in interaction with the GNP molecule (colored in orange). PDB identifier: 5P21. (b) Structure of Ras in complex with the RBD and CRD domains of Raf (colored in wheat). The GNP molecule is again colored in orange. PDB identifier: 6XI7.
(PDF)

**S1 Table. Identifying functionally important sites in natural protein families: Detailed results.** The first two columns present the name of natural protein families that are considered and the corresponding label used in Fig 6 and S11 Fig. Next, the 'DMS' column indicates the shape of the DMS data, 1 for unimodal and 2 for bimodal. The symmetrized AUC columns correspond to the results shown in Fig 6—conservation is abbreviated by 'Cons.'. The best score among the methods is highlighted in bold for each family. Finally, the number of true positive sites found both by ICOD and by conservation ('Both'), by ICOD only ('ICOD') and by conservation only ('Cons.') are given. The last columns provide the size (or length) $L_S$ of the sector and the protein length $L$.
(PDF)

**S2 Table. Diversity and depths of MSAs.** The first column gives the name of the natural families considered. The second and the third columns give the mean Hamming distance between two sequences of the MSA, respectively for the smallest and largest phylogenetic cutoffs considered ("$C_{min}$", "$C_{max}$"). Phylogenetic cutoffs are defined and their values are given in the paragraph "MSA construction" of the Methods section. The average value over all families of the mean Hamming distances for "$C_{min}$" and "$C_{max}$" are respectively 0.19 and 0.59. The fourth and the fifth columns give the depth of the MSA, again for the two extreme cutoffs. Finally, the sixth and the seventh columns give the effective depth [7, 77] of the MSA (where the threshold of Hamming distance below which neighbouring sequences are lumped together [7, 77] is set to 0.2), again for the two extreme cutoffs.
(PDF)

## Acknowledgments

A.-F. B. thanks Ned S. Wingreen and Shou-Wen Wang for inspiring discussions.

## Author Contributions

**Conceptualization:** Nicola Dietler, Anne-Florence Bitbol.

**Formal analysis:** Nicola Dietler, Alia Abbara, Anne-Florence Bitbol.

**Funding acquisition:** Anne-Florence Bitbol.

**Investigation:** Nicola Dietler, Alia Abbara, Subham Choudhury, Anne-Florence Bitbol.

**Methodology:** Nicola Dietler, Alia Abbara, Anne-Florence Bitbol.

**Software:** Nicola Dietler.

**Supervision:** Anne-Florence Bitbol.

**Visualization:** Nicola Dietler.

**Writing – original draft:** Nicola Dietler.

**Writing – review & editing:** Anne-Florence Bitbol.

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
