## [Decision Letter · Decision Letter 0]

13 Jun 2024

Dear Dr. Bitbol,

Thank you very much for submitting your manuscript "Impact of phylogeny on the inference of functional sectors from protein sequence data" for consideration at PLOS Computational Biology.

As with all papers reviewed by the journal, your manuscript was reviewed by members of the editorial board and by several independent reviewers. In light of the reviews (below this email), we would like to invite the resubmission of a significantly-revised version that takes into account the reviewers' comments.

All reviewers agree on the high scientific quality of the paper, but raise a number of questions, including important issues like the quality and conciseness of the presentation (e.g. accuracy and completeness of the methodological part, overall length of the manuscript), and the robustness of the results (e.g. phylogenetic resampling, phylogenetic tree structure, pseudocounts etc). These comments seem substantial, and should be easily addressable by the authors.

We cannot make any decision about publication until we have seen the revised manuscript and your response to the reviewers' comments. Your revised manuscript is also likely to be sent to reviewers for further evaluation.

Sincerely,

Martin Weigt

Guest Editor

PLOS Computational Biology

Nir Ben-Tal

Section Editor

PLOS Computational Biology

All reviewers agree on the high scientific quality of the paper, but raise questions a number of questions, including important issues like the quality and conciseness of the presentation (e.g. accuracy and completeness of the methodological part, overall length of the manuscript), and the robustness of the results (e.g. phylogenetic resampling, phylogenetic tree structure, pseudocounts etc). These comments seem substantial, and should be easily addressable by the authors.

Reviewer's Responses to Questions

**Comments to the Authors:**

Reviewer #1: I have attached a file with my comments.

Reviewer #2: This paper explores the impact of phylogeny on the inference of protein sectors using different methods: ICOD, covariance, conservation, and SCA. For this, the authors conducted an exhaustive study on synthetic data generated by a minimal model of sequences with functional sectors and a phylogenetic tree. The results enabled the dissection of phylogeny's role in selection signals and the establishment of a method hierarchy based on phylogeny robustness for sector identification and mutational effect inference. The insights derived from synthetic data analysis helped to interpret important site identification in natural protein families using the methods under study.

The authors address the problem rigorously, and their main result is that predictions with ICOD and conservation methods are more accurate due to their robustness to phylogeny. Although their work is sound and convincing, a few points require further clarification.

1- Conclusions derived from Figure 3 on the impact of phylogeny in matrices spectra are based on the observation or not of contrast of the outlier eigenvalue with the rest of the spectrum. However, this is done for a single simulation of the evolutionary process; how robust is this to fluctuations due to phylogenetic sampling? For instance, it is not clear why, if phylogeny impacts large eigenvalues of the covariance matrix, the contrast of its smaller eigenvalue with the rest of the spectrum decreases for strong phylogeny. Is this just sampling noise? 

2- The description of Figure 6 does not explain why the recovery decreases for the covariance matrix when selection strength increases for \\tau*=140 without the presence of phylogeny. I guess this is because increased conservation and covariance are susceptible to it in contrast with the ICOD method; however, it would be recommended to explain this in the text for completeness. 

3- In the discussion section of the article, it is suggested that the superiority of the conservation method over SCA in the analysis of natural protein families is a result of its robustness to phylogenetic correlations. However, for synthetic data, it was found that this is only possible under strong phylogenetic regimes, which is unlikely in the case of homologous protein families and, in particular, for all the families studied. Could the authors elaborate on this hypothesis?

4- A primary result of this work is the demonstration that the ICOD method is consistently more robust than other methods in the presence of correlations of phylogenetic origin. However, from the manuscript, it is understood that pseudocounts regularization is used only for the ICOD method during covariance matrix calculation, while for other methods, it is not. Pseudocount is a regularization scheme developed to correct for non-invertible matrices due to finite sampling, an effect to which phylogenetic correlations also contribute by reducing the effective number of sequences. At the same time, pseudocount has been shown to improve the predictions of other methods that do not necessarily require the inversion of the covariance matrix but use the information it contains. If the above is correct, is it possible that pseudocount is a way to reduce the bias of phylogenetic correlations, and therein lies an advantage over the other methods? Could the authors verify that this is not what makes the difference?

5- During the synthetic data study, a binary, symmetric, and homogeneous tree was considered. However, the spectrum of eigenvalues of the covariance matrix is known to be modified for more realistic phylogenies, although a power law tail of large eigenvalues is maintained. Could the authors discuss the effects of this simplification on the paper's conclusions and how it would impact a typical tree of natural data, which is inhomogeneous and has multiple branching events?

Reviewer #3: Attached as a pdf

**Have the authors made all data and (if applicable) computational code underlying the findings in their manuscript fully available?**

Reviewer #1: Yes

Reviewer #2: Yes

Reviewer #3: Yes

PLOS authors have the option to publish the peer review history of their article (what does this mean?). If published, this will include your full peer review and any attached files.

Reviewer #1: No

Reviewer #2: No

Reviewer #3: No
---

## [Decision Letter · Decision Letter 1]

10 Sep 2024

Dear Dr. Bitbol,

We are pleased to inform you that your manuscript 'Impact of phylogeny on the inference of functional sectors from protein sequence data' has been provisionally accepted for publication in PLOS Computational Biology.

Best regards,

Martin Weigt

Guest Editor

PLOS Computational Biology

Nir Ben-Tal

Section Editor

PLOS Computational Biology

Reviewer's Responses to Questions

**Comments to the Authors:**

Reviewer #2: I want to thank the authors for their thorough and thoughtful responses to the points raised. The revisions have significantly enhanced the document's quality, making it a highly engaging and valuable work.

Reviewer #3: I thank the authors for giving detailed responses to my questions and taking into account my comments on the manuscript.

**Have the authors made all data and (if applicable) computational code underlying the findings in their manuscript fully available?**

Reviewer #2: Yes

Reviewer #3: Yes

PLOS authors have the option to publish the peer review history of their article (what does this mean?). If published, this will include your full peer review and any attached files.

Reviewer #2: No

Reviewer #3: No

---

## [Editor Report · Acceptance letter]

16 Sep 2024

PCOMPBIOL-D-24-00658R1 

Impact of phylogeny on the inference of functional sectors from protein sequence data

Dear Dr Bitbol,

I am pleased to inform you that your manuscript has been formally accepted for publication in PLOS Computational Biology. Your manuscript is now with our production department and you will be notified of the publication date in due course.

With kind regards,

Anita Estes
